



# Are precipitation anomalies associated with aerosol variations over Eastern China?

Xiangde Xu[1], Xueliang Guo[1,2,*], Tianliang Zhao[3,*], Xingqin An[1] ,Yang Zhao[1], Jiannong Quan[4], Fei Mao[1], Yang Gao[4], Xinghong Cheng[1], Wenhui Zhu[1], and Yinjun Wang[1]

[1]State Key Laboratory of Severe Weather (LASW), Chinese Academy of Meteorological Sciences, Beijing 100081, China,

[2]Key Laboratory for Cloud Physics, Chinese Academy of Meteorological Sciences, Beijing 100081, China , [3]Collaborative

[3]Innovation Center on Forecast and Evaluation of Meteorological Disasters, Key Laboratory for Aerosol-Cloud-Precipitation

of China Meteorological Administration, Nanjing University of Information Science & Technology, Nanjing 210044,China,

[4]Beijing Weather Modification Office, Beijing 100089, China

*Correspondence to*: Tianliang Zhao (lzhao@nuist.edu.cn)

**Abstracts**. In Eastern China (EC), strong anthropogenic emissions deteriorate the atmospheric environment harbored by the upstream Tibetan and Loess Plateaus, building a south-north zonal distribution of high anthropogenic aerosols. This research analyzed the interannual variability of precipitations with different intensities in the EC region from 1961 to 2010. We found that the frequency of light rain significantly decreased and the occurrence of rainstorm, especially the extraordinary rainstorm significantly increased over the recent decades. The extreme precipitation events presented the same interannual variability pattern with the frequent haze events. Moreover, the extreme rainfall events of various intensities showed a regular interannual variability trend. During the 1980s, the regional precipitation trends in EC showed an obvious "transform" from more light rain to more extreme rainstorms. The running correlation analysis of interdecadal variation further verified that the correlation between the increasing aerosol emissions and the frequency of abnormal precipitation events tended to be more significant in the EC. The correlation between atmospheric visibility and low cloud amounts, which are both closely related with aerosol concentrations, had a spatial distribution of "northern positive and southern negative" pattern, and the spatial distribution of the frequency variability of regional rainstorms was "southern positive and northern negative". After the 1990s, the visibility in summer season deteriorated more remarkably than other seasons, and the light rain frequency decreased obviously while the rainstorm and extraordinary heavy rainfall occurred more frequently. There were significant differences in the interdecadal variation trends in light rain and rainstorm events between the high aerosol concentration areas in the EC and the relatively "clean area" in western China. The aircraft measurements over the EC confirmed that the diameters of cloud droplets decreased under high aerosol concentration condition, thereby inhibiting weak precipitation process.





## 1. Introduction

It is widely acknowledged that the global mean temperature has been increasing since 1850 and will continue to increase in the following decades (IPCC, 2007). The long-term forcing of a warming environment might change precipitation, the regional and global water cycle (Allan and Soden, 2008; Allan and Ingram, 2002). The heavy precipitation events showed a overall increasing trend as the result of global warming (Allan and Ingram, 2002; Trenberth et al., 2003). Since the 1950s, the precipitation has increased remarkably at high latitudes, also increased at tropical marine areas, but it underwent some decrease at tropical mainland (New et al., 2001; Kumar et al., 2006; Bosilvoich et al., 2005).

The variation of aerosols in atmosphere is also an important factor to influence the water cycle in regions where bearing long-term high aerosol loading (Ramanathan et al., 2005; Ramanathan et al., 2007; Koren et al., 2008; Levin and Cotton, 2009; Li et al., 2011). Under the background of global warming, the regional precipitation tends to have more complex temporal and spatial distribution patterns. The variations of precipitation could be reflected by the different-grade precipitation, and even by frequency changes of extreme precipitation events (Lau and Wu, 2007), which could threaten the social economy and is seriously concerned by governments, public and scientific community.

Precipitation is not only influenced by atmospheric circulation structure related with land-sea discrepancy and land-sea water vapor exchange, but also by local cloud microphysical processes (e.g., CCN, IN). Studies have shown that atmospheric aerosols might add cloud droplets number concentrations (CDNC), and change cloud lifetime, and restrain or enhance precipitation (Khain et al., 2005; Rosenfeld et al., 2007; Rosenfeld and Coauthors, 2008; Stevens and Feingold, 2009; Fan et al., 2013).

Since the middle 1980s, China experienced a rapid development in industry and agriculture. As a result, a huge amount of industrial emissions and biomass burning significantly released particulate matters into the atmosphere. The study shows that there was no obvious change in annual precipitation in China, but the extremely heavy rainfall area, mainly in the EC, had expanded (Zhai et al., 1999). However, the regional annual precipitation, summer precipitation, and extreme precipitation events had an obvious rising tendencies in middle and lower Yangtz River Basin of EC (Wang and Zhou, 2005). The numerical simulations also presented that the increase of aerosols could decrease the summer convective precipitation in the intensity under 30 mm h$^{-1}$, and increase summer strong convective precipitation in the rates above 30 mm h$^{-1}$ in China (Guo et al., 2014). With a rapid increase of aerosols, not only local light rain over wide areas could decrease, but also local extremely heavy rain could be triggered, inducing frequent flood (Guo et al., 2014; Fan et al., 2015).Light rain tended to decrease and at the same time the extremely heavy precipitation had increasing tendency in the EC (Choi et al., 2008; Qian et al., 2007; Qian et al., 2009). This phenomena might be the strong signal of climate variability connecting to global warming together with the increased emissions of anthropogenic aerosols.

Aerosols might also change Asian monsoon system (Bollasina et al., 2011). There are great uncertainties in the interactions





between the internal influence factors including complex influences of land-sea discrepancy, aerosols and cloud and
precipitation processes of Asia monsoon systemand external forcing factors.
The large amounts of anthropogenic aerosols not only deteriorate the environment over large spatial scales, but they might
induce the rapid change of regional climate and water cycle. In addition, the high aerosol concentration zone in the
north-south direction over China is located on the eastern side of the Tibetan plateau and the Loess plateau, which might be
connected with the leeward slopes of the large terrain structure of the plateaus. The regions from the plateaus to EC may be
an ideal place to identify the climate forcing of aerosols with following questions: comparing the   interannual variation
trends in various precipitation intensities betweenthe polluted EC with the clean region over the Tibetan plateau.The
previous investigations of this issue primarily focused on limited cases. The climatic forcing of aerosols on precipitation
extremes in large-scale continent region and its physical causes remain uncertain. By using precipitation and visibility data in
a 50-year period and satellite, aircraft and surface aerosol observational data in recent years in China, the climatic impacts of
aerosols on interannual variability of various precipitation intensities and their physical links were investigated in this study.
**2. Data**
In this work, we adopted annual average AOD data in 2001-2010 from Moderate Resolution Imaging Spectroradiometer
(MODIS); monthly frequency of different-intensity of precipitation events including extraordinary storm ( >200mm, the
precipitation intensity classification standard for 24 hours), rainstorm (100-200mm), large rainstorm heavy rain (25-50 mm),
moderate rain (10-25mm), light rain (0.1-10mm) from 601 stations in China over 1961-2010 (Datasets from the National
Meteorological Information Center of China Meteorological Administration); In addition, the meteorological and
environmental data including monthly haze days of 2513 stations, daily visibility of 598 stations and daily low cloud cover
of 753 stations in China as well as the daily $PM_{2.5}$ data of 946 stations in 2013-2014  in China were also used.
In order to analyze the regional variations in aerosols over Eastern China, we adopt the equivalent visibility by excluding the
influence of natural factors (Rosenfeld et al., 2007) on the observed visibility based on the meteorological data observed
from 598 stations in 1961-2010 were used in this study. The equivalent visibility was corrected based on the the
followingformula (1) under the relative humidity from 40% to 99%.
$$\frac{\text{VIS}}{\text{VIS(dry)}} = 0.26 + 0.4285\log(100 - \text{RH}) \tag{1}$$
The characteristics of aerosol and cloud droplets size were comprehensively analyzed based on the aerosol-cloud data
obtained from aircraft flights carried out in Beijing and its surrounding regions during 2008-2010 by the Beijing Weather
Modification Office. The scientific detection time was from May to August during 2008-2010.The observed clouds were
mainly stratus cloud, stratocumulus and cumulus clouds, and the maximum detection altitude was 7000 m. There were 40
flights carried out during the experiment period. Aircraft measurements were usually carried out within 2-6 h before the





clouds precipitated. The flight area and tracks were shown in Fig. 1. The Passive Cavity Aerosol Spectrometer Probe
(PCASP-200,DMT Co.) was used for observing aerosol particle size in 0.1-0.3μm. The probe of Cloud, Aerosol and
Precipitation Spectrometer (CAPS,DMT Co.) was used for observing cloud droplets in 0.6-50μm. The probes were returned
to the DMT for standard calibration before starting measurements in each year. In addition, the probes were calibrated using
the spheres of polystyrene latex (PSL) of Duke Scientific Corporation for each month. Considering the influence of cloud
droplets on aerosol probing, the averaged aerosol concentration below 300m of cloud base was calculated to represent
aerosol concentration in clouds. The cloud droplet   measurements were made within clouds at 100m height intervals. The
data were processed into two or more samples when the clouds were multiply layered.

**3 Haze distributions in Eastern China harbored by large plteaus**

Due to the influence of the terrain on the typical westerly winds in Eastern China, the air flowing from the windward
plateaus descends in a north-south oriented zone between about 110ᵒE and 125ᵒE (upper panel of Fig. 2). Accompanying this
strong downward current are weak winds in the near-surface layers in the lee side of the plateaus. These air flow and wind
condition lead to development of a "harbor" accumulating air pollutants in EC. The weak wind and downward current areas
coincide well with the centers of frequent haze events in China (lower panel of Fig. 2). The "susceptible region" of haze
events over Eastern China from the eastern edge of the plateaus to the lower flatlands is associated with the "harbor" effect
of the unique topography under specific meteorological conditions that trap air pollutants (Xu etl al., 2016).

**4. Change trends in various precipitation intensities**

The interannual variation trends of extreme precipitation events with various intensities including light rain, moderate rain,
heavy rain, rainstorm, the large rainstorm, extraordinary rainstorm (Fig. 3a) were comparatively analyzed and it is found that
the interannual variation trend differences for the six various precipitation intensities were significant. The light rain
frequency trend significantly decreased (Fig. 3a, A), the moderate rain frequency trend slightly declined (Fig. 3a, B), the
interannual change trend of heavy rain frequency was not significant (Fig. 3a, C), the rainstorm and large rainstorm events
increased significantly (Fig. 3a, D, E, F) in EC. Especially since 1980s, the extremely heavy precipitation events have
become more frequent, showing an obvious transforming characteristics of frequent heavy rain and torrential rain. Large
rainstorm and especially large rainstorm extreme events presented significantly an increased trend, along with the frequent
haze weather in EC. Overall rainstorm extreme events were on the rise trend, , but light rain tended to decline significantly.
In contrast,  stations in the Tibetan Plateau (at height of >4000m), a relative clean area in China, were selected for a
statistical analysis of interannual variation trend of light rain frequency, indicating that the characteristic of the decreased
trend of light rain frequency was not significant in the Tibetan Plateau (Fig. 3b,A).



**5. Distribution of frequency of extreme rainstorm events and visibility**
The negative variability area of light rain frequency almost matched with positive variability of visibility and haze frequency
in EC (Fig. 4a,b and c), which are well consistent with the area of high aerosol concentrations and frequent haze events (Fig.
2a,b). The light rain frequency reduction in China was closely associated with the enhancement of aerosol levels in the
atmosphere (Qian et al., 2009).
It is noteworthy that the negative trend areas of light rain almost covered a large part of areas in China and all eastern China
(Fig. 4c). This might be closely related with temporal and spatial variation trends of summer monsoon activity which offered
a suitable dynamic background for the effect of aerosols on clouds and precipitation.Figure 5a shows that the spatial
distribution of the rainstorm frequency variability was "southern positive and northern negative" in summer during
1961-2010, while the correlations between visibility and low-level cloud amount were distributed with the "northern positive
and southern negative" pattern during 1961-2010 in EC (Fig.5b), indicating that the effect of aerosols on summer
convective precipitation was more obvious in southern part than that in northern part of EC.

There were obvious differences in the interdecadal precipitation decreasing rate of the various precipitation intensities in the
EC region (Fig.6a). In this region, the summer interdecadal precipitation variability revealed the influence features of
aerosols on clouds and precipitation process, that negative variability stations of light rain made up the majority (about
87.6%), the positive variability stations of moderate rain were approximately equal to the negative ones (about 51%), the
positive variability stations of heavy rain (about 71.3%) were much more than the negative ones indicating the reverse trend.
The positive variability stations of rainstorm with daily precipitation >50mm, including catastrophic rainstorm over
100mm occupied obvious majority (about 78.9%). In China, in the recent decades, the rapid increase of the anthropogenic
aerosol particles in the atmosphere may make the cloud droplet number concentration increased but the size of cloud droplet
decreased, thus changing the life time of the cloud and suppressing the precipitation, especially for the light rain (Qian et al.,

144 2009).

As mentioned above, the light rain frequency reduced significantly; the moderate rain frequency changed unobvious; heavy
rain increased relatively obvious, rainstorm and catastrophic rainstorm increased significantly obvious in eastern China,
indicating the obvious change characteristics of regional precipitation from light rain changed to heavy rain and even the
catastrophic rainstorm along with the frequent haze trend in EC.
Although severe precipitation event mainly depended on dynamical and thermodynamic processes and water vapor source in
the atmospheric circulation and deep convective activity, aerosol's "Albrecht effect" considered that increased cloud droplet
concentration and decreased cloud droplet size influenced by aerosol would suppress cloud precipitation process and extend
cloud maintenance time. The extension of the cloud life time might save the potential that triggering the abnormal severe
precipitation extreme events when the cloud droplets coagulation condition was mature. This mechanism could partly





explain the significant light rain reduction trend (Fig. 6a) and the spatial consistency indicating the precipitation rate
transformation characteristics of light rain to heavy rain or severe precipitation extreme events (blue arrows in Fig. 6a) in
eastern China. Moreover, in this research, in order to further comparatively analyze the effects of frequent haze weather in
eastern China on extreme precipitation events, we selected the Tibetan Plateau (west of 110 °E, south of 40 °N), a relative
clean area in western China, as the reference area, and calculated frequency variability trends of light rain and rainstorm in
the three different interdecadal periods (1961-1980, 1971-2000, 1981-2010) in the east and west regions. As could be seen
from Fig. 6b (left), in the three stages, the positive variability of light rain was a declining trend while the negative variability
was an increasing trend in EC, while there were no obvious positive or negative variability trend of light rain and rainstorm
in Tibetan Plateau (double-headed arrows in Fig. 6b right).
**6. interannual variability between atmospheric visibility and precipitation anomalies**
We calculated the correlation coefficients between regional averages of visibility with the frequency of light rain, heavy rain,
extremely torrential rain over EC region (east to 110 °E, south to 40 °N), respectively in consideration of   correlation pattern
between visibility and various precipitation intensities. Therefore, taking summer months (June, July and August) as
examples, the 20-year running correlation coefficients of visibility and precipitation were obtained ( Figs. 7 a, b and c). The
running correlation coefficients curve of visibility and light rain reached the confidence level (90%), and the running
correlation coefficients curve of visibility and heavy rain and extremely heavy rain reached the confidence level (90%). This
statistical analysis results further verified that with the increase of aerosol emissions, visibility and various precipitation
exhibited increasing significant correlation trends (Fig.6).

In order to investigate the seasonal and annual correlation pattern between regional visibility and light rain, as well as
regional visibility and extremely precipitation events in EC, we illustrated the annual cross-section of monthly anomaly of
visibility and light rain, visibility and heavy rain, as well as visibility and extremely heavy rain. Through comprehensive
comparison of Figs. 8 a, b, c and d, we could find significant positive correlation between visibility and light rain, showing
that the poor visibility surpressed light rain frequency. Moreover, there was significant difference between the changing
trend of extremely precipitation events frequency and light precipitation frequency. The change of heavy precipitation
frequency from 1960s to 1980s was not as prominent as that at the latter period of 1990s, during which time visibility
deteriorated remarkably, heavy and extremely heavy rain occurred frequently. Compared to other seasons, the influence
effect of poor summer visibility was more significant in China, showing light rain frequency decreasing significantly and
sudden heavy rain and large heavy rain frequency increasing.
The increased atmospheric aerosol concentration may reduce the solar radiation to surface and decrease surface temperature.





At the same time, the polluted black carbon aerosols can strongly absorb solar short-wave radiation and directly heat
low-level atmosphere, and form a temperature inversion structure (Bollasina et al, 2011; Zhang et al., 2009; Bond et al.,
2013; Bond et al., 2011; Grant et al., 2014; Seinfeld et al., 2008). Therefore, the aerosol-radiation interaction could change
the atmospheric stability and alter local or region atmospheric circulation and precipitation process. This temperature
inversion structure increases the stability of atmospheric boundary layer and provides an important condition for the frequent
occurrence of haze and fog events. The stable low-level structure also inhibits the weak convection development of
atmospheric boundary layer, so as to reduce the formation of low-level clouds and weak precipitation process. However, the
strong dynamic convergence disturbance could destroy the stability of atmospheric boundary layer and cause the formation
and development of severe rainstorms.
To further clarify the relation between aerosol concentration and light rain frequency, the light rain frequency distribution
from 601 stations in July, 2013 is displayed (Fig.9). It shows in Fig. 9 that the light rain events have significantly declined in
the Yangtze River region of EC with high aerosol concentrations have and but enhaced while t in the relative clean region of
Tibetan Plateau.
To reveal the relationship between aerosols and atmospheric vertical thermal structure, the correlation between PM2.5
concentration and atmospheric thermal structure in both polluted and clean areas in July, 2013 was investigated (Fig. 10).
The stations of Changsha and Hongjia located in Hunan and Zhejiang provinces in EC respectively were selected to
represent the less light rain region while those of Linzhi and Dingriin of Tibet were selected to represent the high-frequency
light rain region. The correlation coefficient profiles between the observed surface daily $PM_{2.5}$ concentration and
atmospheric temperature profiles derived from high-resolution L-band sounding were calculated. The correlations at
Changsha and Hongjia stations (Figs.10a-b) show that the correlation between $PM_{2.5}$ and temperature profiles presented an
"inverse phase" characteristic, indicating they were negatively correlated at low boundary layer and positively correlated at
upper boundary layer or troposphere, reflecting the high aerosol concentrations in a thermal structure similar to temperature
inversion layers like "cold at low-layer and warm at upper-layer"in the eastern China. On the contrary, the correlations in
Linzhi and Dingri stations in theTibetan Plateau (Fig. 9c,d) indicate that an unstable atmospheric structure with "warm at
low-layer and cold at upper-layer" for a favorable condition for the occurrence and development of weak convection and
light rain events in the Tibetan region.
**7. Physical connection between aerosols and precipitation**
According to the results of observation and modeling studies, the increased aerosol concentrations could reduced effective
particle radius and increased number concentration of cloud droplets, and latent heat release (Khain et al., 2005; Van den
Heever et al., 2006; Tao et al., 2007; Altaratz et al., 2014).The increase of cloud droplets concentrations would delay





raindrop formation, thereby lessening light precipitation (Qian et al., 2009) leadingthe negative correlation between aerosol
and precipitation in China (Choi et al., 2008).
In order to further investigate the relationship between aerosols and cloud droplets, the observed data collected by aircraft in
north China during 2008-2010 were used. The vertical profiles of cloud droplets data at each level under different aerosol
state obtained by 40 aircraft is shown in Fig. 11a. From Fig. 11a and Fig. 11b, aerosol Albrecht "cloud lifetime effect" was
significant in the Northern EC. As shown in red profile of Fig. 11b, under the background of high aerosol concentration, the
cloud droplets sizes were smaller and increased slowly with the increasing altitude. In addition, from the cloud base to
2000m, the cloud droplets size remained less than 20 microns (Fig.11b), resulting in precipitation delay, in favour of cloud
system development to form heavy rain easily. In addition, cloud droplets diameter enlarged quickly with the increase of
height, and reached 30 microns easily to forming light rain at 1000m altitude under low aerosol concentration (green profile
in Fig.11b). The above aircraft observation analysis showed that under the condition of insufficient water vapor in the North
China relative to that in the southern China, high aerosol concentration could reduce cloud droplets size, increase cloud
droplets concentration, extend cloud lifetime, and this would restrain the development of low clouds, especially restrict the
light rain process.
**8. Discussion and conclusions**
Aerosols have complicated effects on clouds and precipitation, depending on many factors such as aerosol properties,
topography and meteorological conditions. The most previous investigations of aerosol impacts on clouds and precipitation
are primarily based on limited cases in relatively smaller spatial and temporal scales. the climate forcing of aerosols on
precipitation in large-scale continent region and physical causes remain uncertain. By using precipitation and visibility data
in 50 years and satellite, aircraft and surface aerosol data in recent years in China, the impacts of aerosol variations on
interannual variability of various precipitation intensities of precipitation events and their physical causes are investigated.
Accompanied with the frequent haze events in EC, the light rain frequency trend significantly decreased. Especially, since
the 1980s the extremely heavy precipitation event have occurred   more frequent withan obvious transform from light rain to
frequent heavy rain and rainstorm. From 1960s to 1970s, the monthly visibility and light rain presented a significantly
positive correlation, while the visibility was in good condition, and the light rain frequency was also in high value. In recent
20 years, the dramatically increased aerosols resulted in poor visibility, and the light rain frequency decreased obviously, and,
heavy and extremely heavy rain occurred more frequently.
The investigation of relation between aerosol concentrations and light rain frequency distributions from 601 stations in July,
2013 in China shows that that the light rain in the Yangtze River region of EC with high aerosol concentration appeared
significantly low-frequency while that in the relative clean region of Tibetan plateau presented significantly high-frequency.





The physical cause of this relation was investigated, and found that the high aerosol concentration was strongly correlated to
the low-level atmospheric warming which tended to form a stable structure that suppressed the occurrence and development
of weak convection and light rain events in eastern China, while this was not be found in the relatively clean region over the
Tibetan plateau.
The findings from this study have some important implications: the frequent haze events in EC not only cause regional
environment deterioration, but also threaten the social economy and people life in large spatial and temporal scales, and
possible induce the long-term change of regional water cycle. This may exacerbate the effect of climate change.

**Acknowledgements**
The study was supported by the National Key R & D Program Pilot Projects of China (JFYS2016ZY01002213;
2016YFC0203304), the National Natural ScienceFoundation of China (91544109; 91644223 ), the project of Environmental
Protection (HY14093355; 201509001) in the Public Interest and Chinese Third Tibetan Plateau Atmospheric Experiment
(GYHY201406001).

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





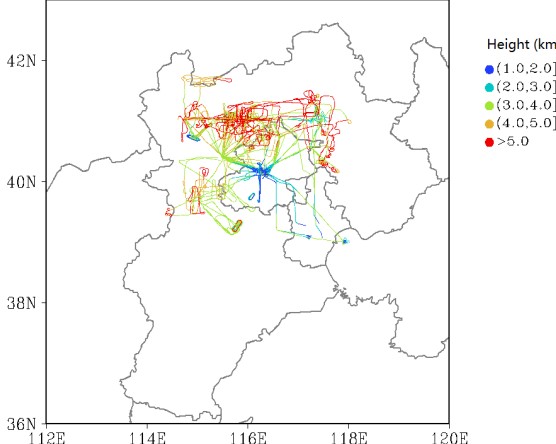


**Figure 1 Area and tracks of 40 aircraft flights carried out in Beijing and its surrounding regions during aerosol-cloud experiment**

**from 2008 to 2010 by the Beijing Weather Modification Office**

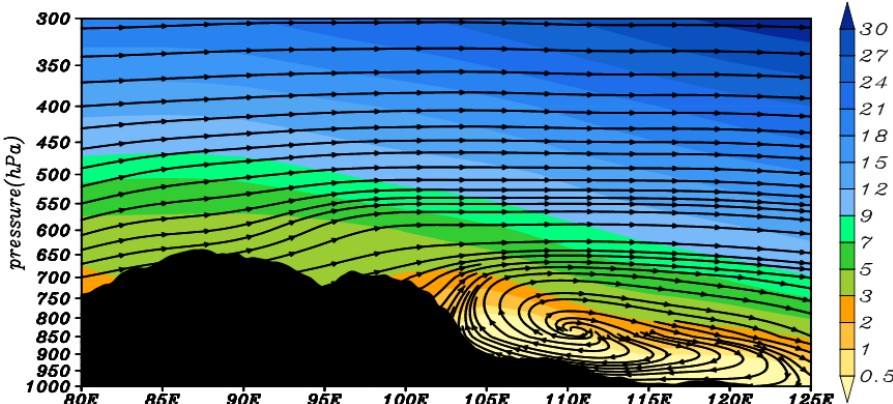

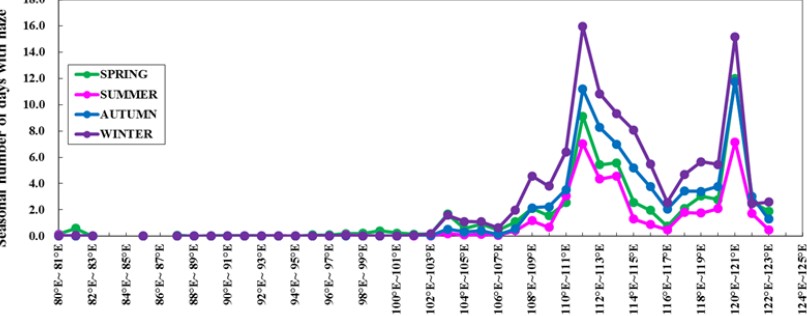

**Figure 2. Cross sections of vertical circulations illustrated by stream lines (upper panel) with the horizontal wind speed (m s-1;**

**color contours) and zonal variations of annual haze event frequency (lower panel) at 27°N-41°N averaged in spring, summer,**

**autumn and winter over 1961-2012. Note that near-surface vertical and horizontal winds are not illustrated well here due to**

**north-south variations in the terrain and approximation of the location of the plateaus (black shaded area) in upper panel. All**

**fields are for the annual-averages.**



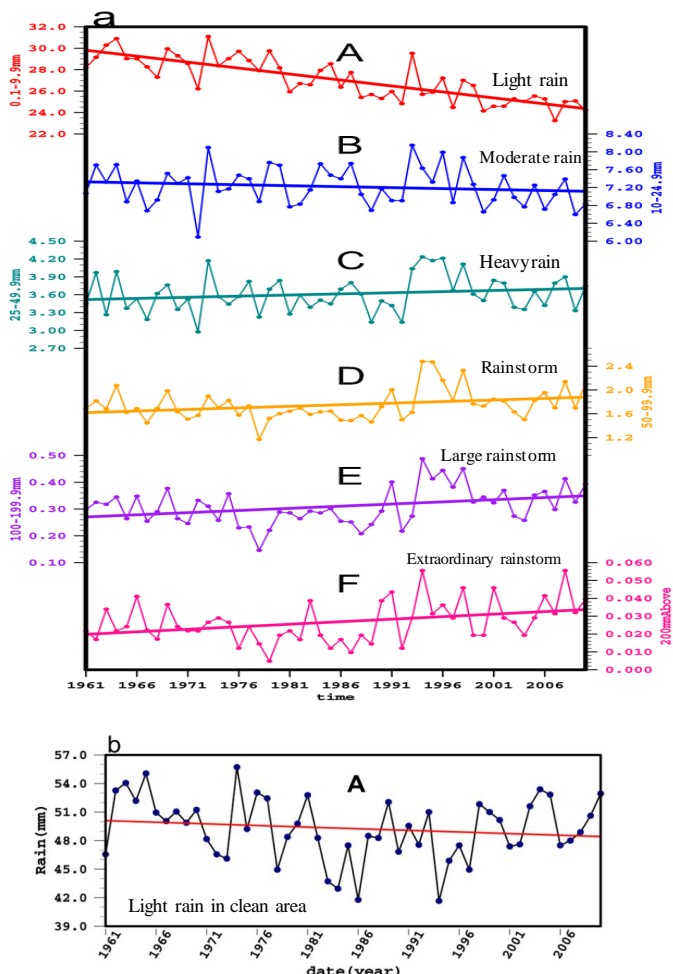

**Figure 3 Interannual variation trends in (a) various precipitation intensities in the high aerosol concentration area in eastern**
**China (east of 110° E) and (b) light rain in relative clean area of Qinghai Tibet Plateau;Note: Various precipitation intensities**
**included light rain (A), moderate rain (B), heavy rain (C), rainstorm (D), large rainstorm (E), and extraordinary storm (F).**

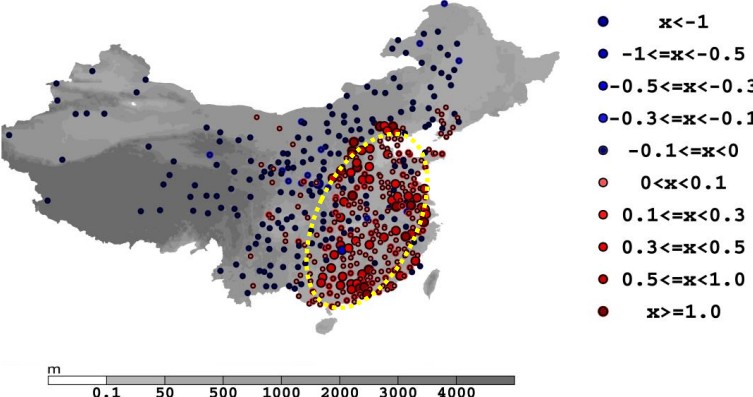






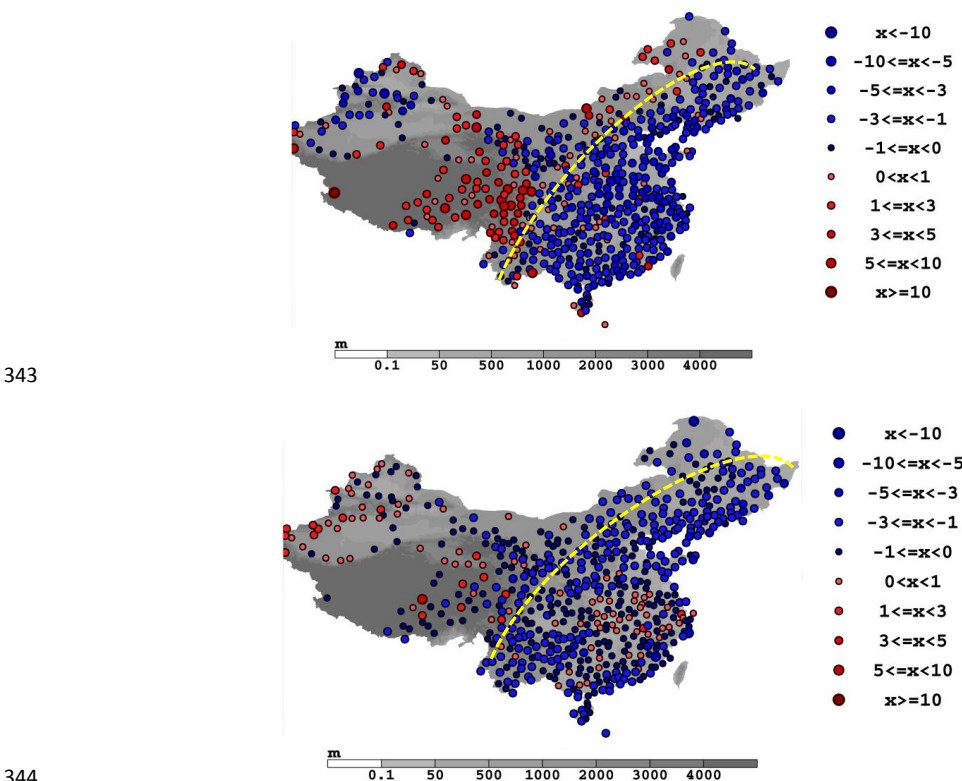



**Figure 4 Interannual variability distribution of haze frequency (a), visibility (b) and light rain frequency (c) in summer in mainland China in 1961-2010.**



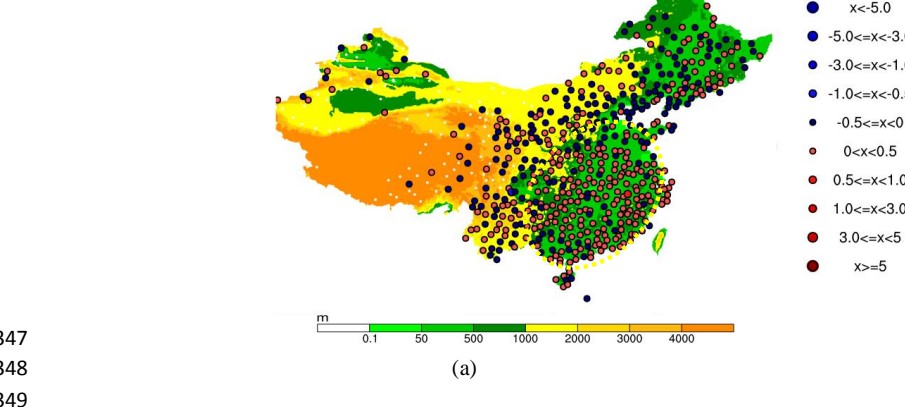


348                                                              (a)






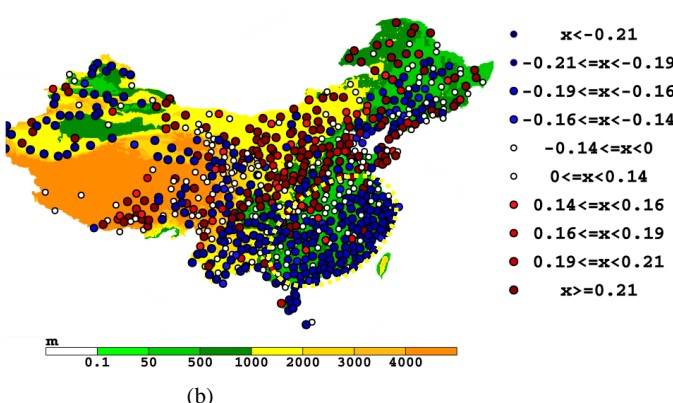


(b)
**Figure 5 The spatial distributions of (a) trends in summertime rainstorm frequency over 1961-2010 in China and (b) correlation**
**coefficients between visibility and low cloud amount in summer of 1961-2010.**

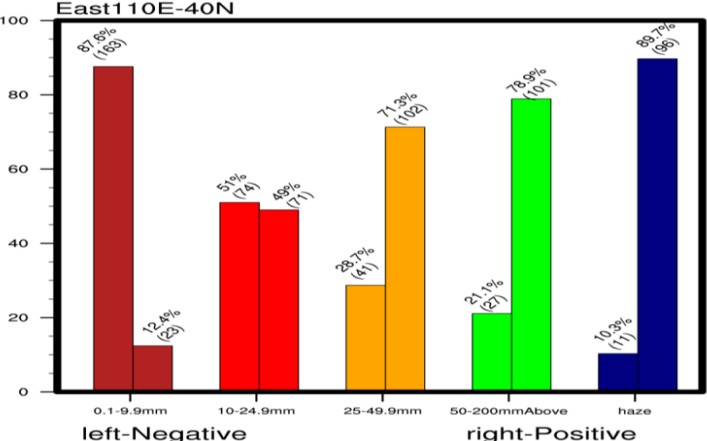


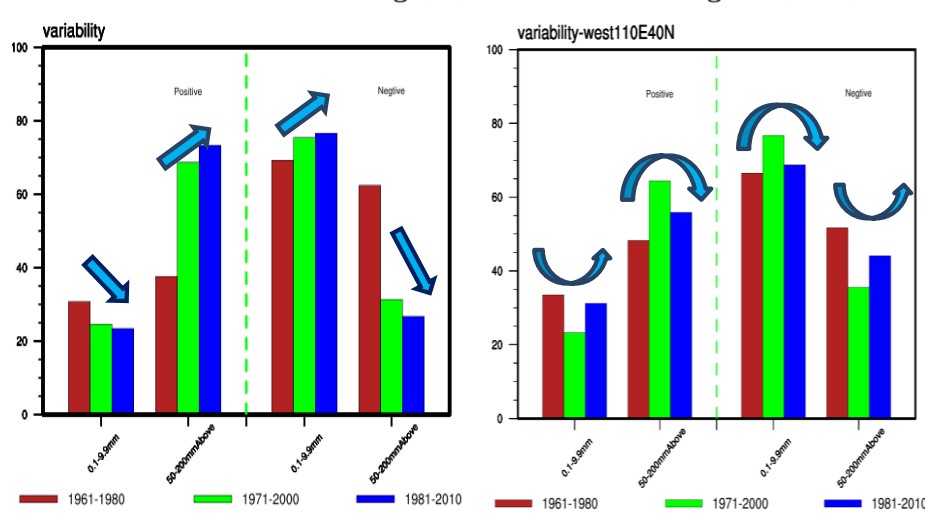


**Figure 6 The ratio of summer haze days and the stations covered by positive and negative variability at various precipitation**
**grades from 1961 to 2010 over in east of 110 °E, south of 40 °N (a), west of 110 °E, south of 40 °N (b). 601 precipitation stations and**
**598 visibility stations in eastern China. The right side is the positive variability and left side is the negative variability)**





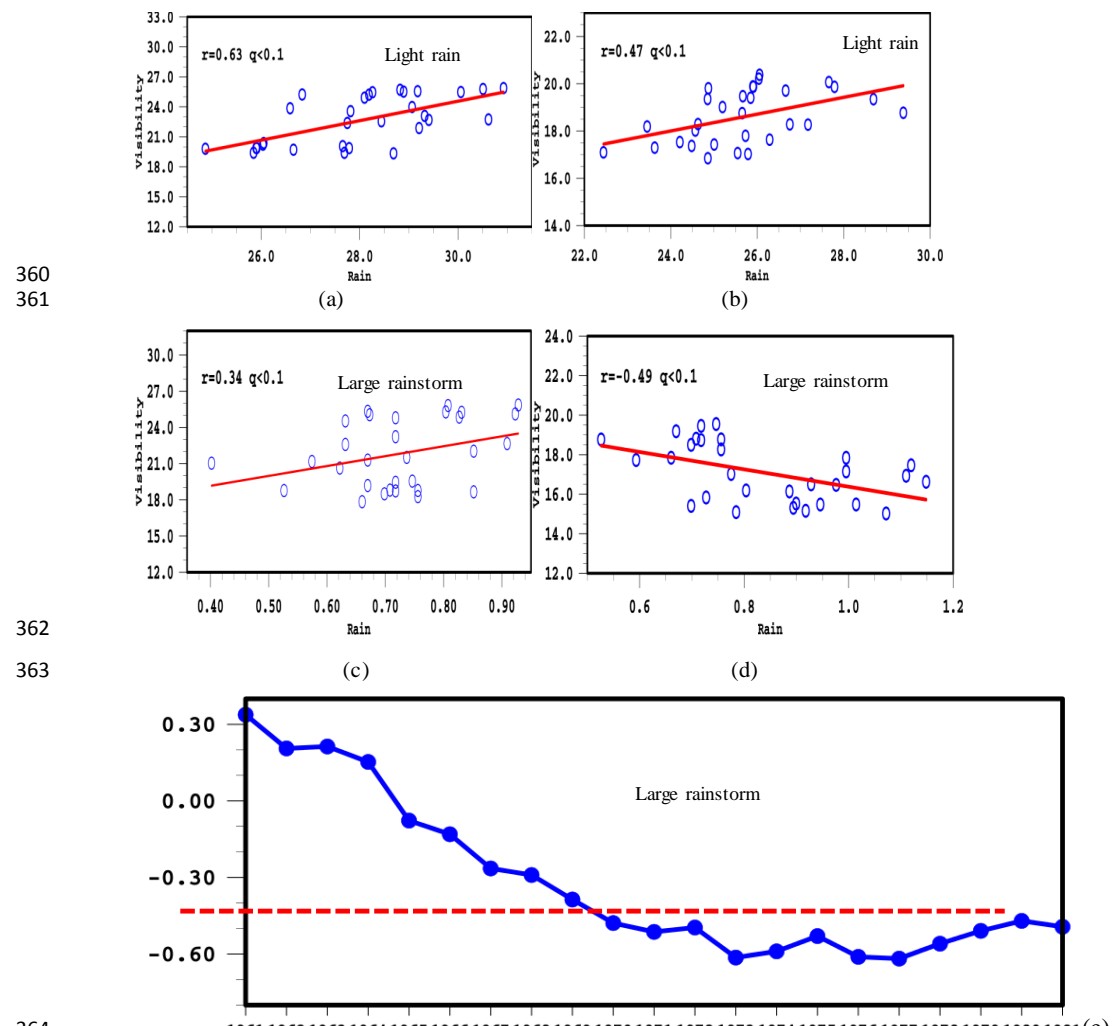

**Figure 7 Correlation between summer average visibility (June, July and August) and light rain frequency in 1961-1990 (a), light rain frequency in 1981-2010 (b), extremely heavy rain event frequency in 1961-1990 (c), extreme heavy rain event frequency in 1981-2010 (d) and the 20-year running correlation coefficients of visibility and precipitation (e), over eastern China (east to 110 °E, south to 40 °N)**

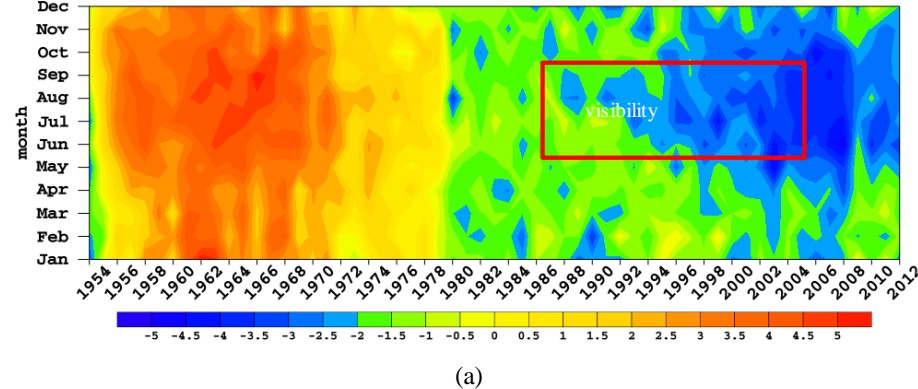

(a)



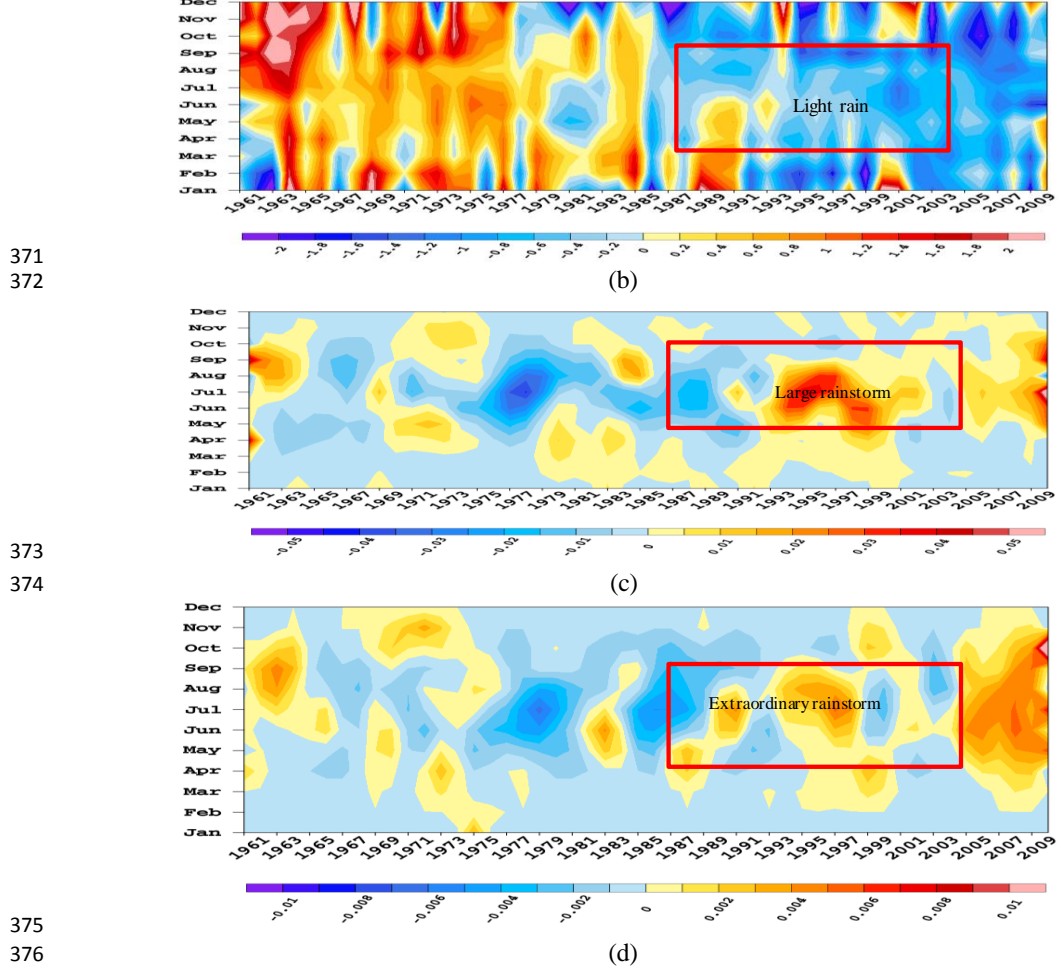

        (b)

        (c)

        (d)


**Figure 8 Annual cross-section of monthly anomaly of visibility (a), light rain (b), heavy rain (c), and extremely heavy rain (d)**

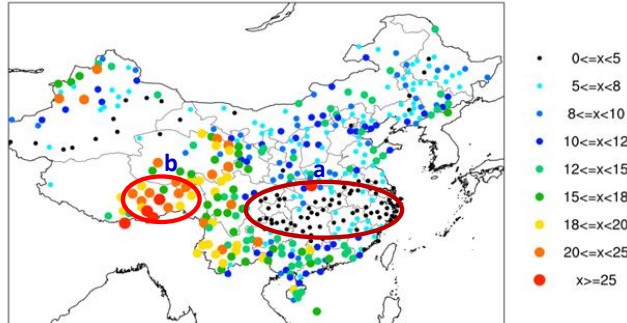


**Figure 9 Light rain frequency distribution of 601 stations in China in July of 2013. The circled region on right is the low-frequency**
**light rain region in middle and downstream region of Yangtze River in eastern China, and that on left is the high-frequency light**
**rain region in relative clean region over Tibet plateau.**



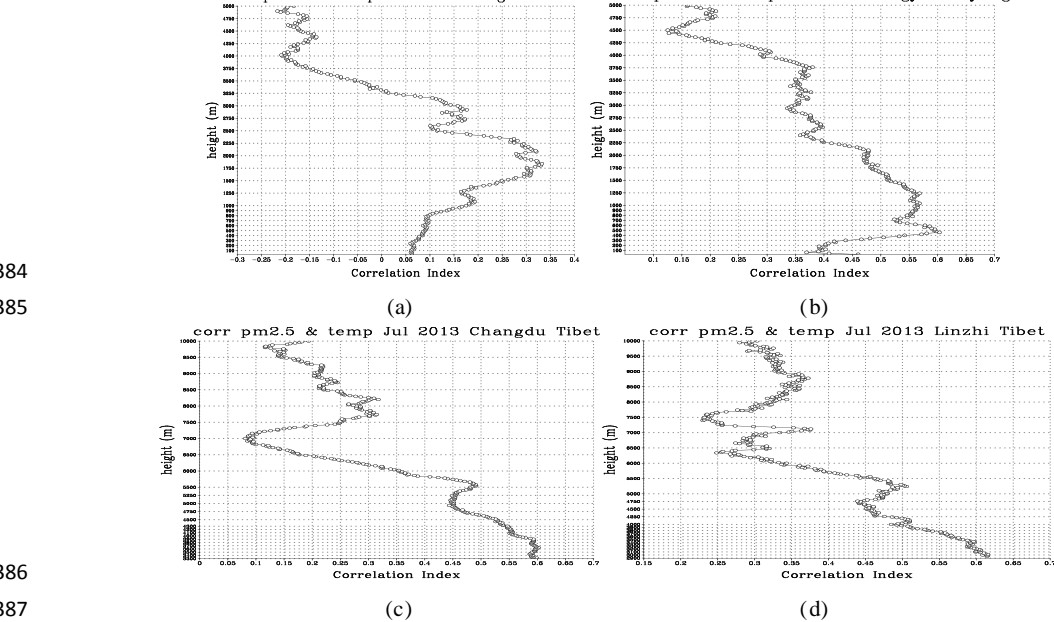


385           (a)                           (b)


387           (c)                           (d)

**Figure 10 Correlation coefficient profiles between the daily PM$_{2.5}$ concentration (12 hour intervals) and atmospheric temperature**
**at different vertical layer from L-band sounding for representing low-frequency light rain regions at stations of Changsha, Hunan**
**(a) and Hongjia, Zhejiang (b), and for representing high-frequency light rain regions at relative "clean area" at stations of Linzhi**
**(c) and Dingri (d) of Qinghai Tibet Plateau.**

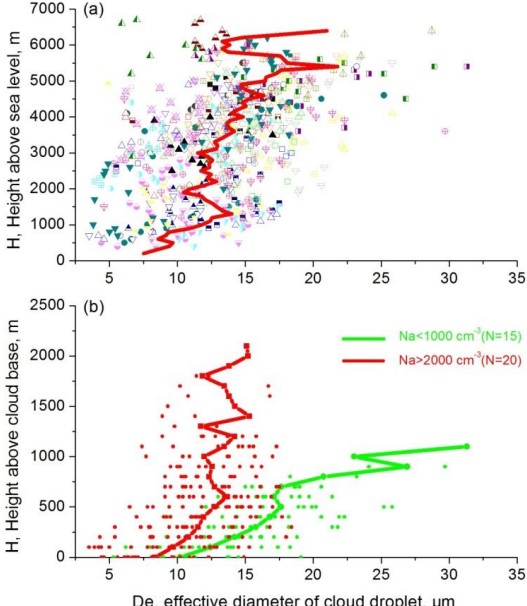


**Figure 11 The vertical profiles of sampling cloud droplets data at each level under different aerosol state(a) and concentration(b)**
**detected by 40 aircraft(Green profile: cloud droplets diameter under low concentration; Red profile: cloud droplets diameter**



**under high concentration).**