# Peer review of "Are precipitation anomalies associated with aerosol variations over 5 Eastern China?"

_Atmospheric Chemistry and Physics, 2016_

## Referee Comment (RC1) · Anonymous Referee #2 · 8 Feb 2017

Summary

The present work investigates how the interannual variability of precipitations with different intensities in the EC region (Eastern China) from 1961 to 2010. The motivation of the work is established in the introduction. Satellite data, Meteorological station observations, and aircraft flight data are used and subsequently introduced with select relevant details. In the main part of the work the results are presented and analyzed, and the authors carefully quantify and discuss the frequency of light rain significantly decreased and the occurrence of rainstorm, especially the extraordinary rainstorm significantly increased over the recent decades. The work addresses an interesting question - Are precipitation anomalies associated with aerosol variations over Eastern China? The key insight is that the aircraft measurements over the EC confirmed that the diameters of cloud droplets decreased under high aerosol concentration condition,

thereby inhibiting weak precipitation process.

The manuscript is thorough, clear, compelling, very well written, and presents the results with good figures and tables. I recommend publication after attending to the following detailed comments.

Detailed comments

Line 71: Please give references for the previous investigations of this issue primarily focused on limited cases (references).

Section 2: Please give a website or reference for MODIS data.

For all of Chinese map, if the author can use LambertEqualArea projection (http://ncl.ucar.edu/Applications/maponly.shtml), that would be nice.

Fig. 10: Please restrict the four panels in the same size.

---

## Referee Comment (RC2) · Anonymous Referee #1 · 10 Feb 2017

Review of "Are precipitation anomalies associated with aerosol variations over Eastern China?" by Xu et al.

General:

This is a comprehensive study on the precipitation-aerosol relationship over a large region (eastern China) for a long period of time (1961-2010). This work involves processing of large amounts of datasets relating to precipitation and aerosols from ground and satellite observations, including monthly haze data at 2513 stations, daily visibility data at 598 stations, and daily cloud cover data at 753 stations, daily $PM_{2.5}$ data at 946 stations and MODIS satellite data. In addition, aircraft measurements are also included and analysed. There are similar studies for different periods and different regions. A study that covers such a large region for such a long period of time is rare.

Overall, this study is well done with thorough analysis. The results are reliable and solid. This study has made a new contribution to our understanding of precipitation-aerosol relationship and the implications of this study should not be underestimated.

The paper can be improved with more in-depth discussion. In the introduction, the authors can write a more comprehensive literature review on the precipitation-aerosol relationship, such as, what are controversy issues, what are the research gaps and what are the possible underlying mechanisms for various processes. Then the review can lead to what the authors will address in this paper. The conclusions and abstract seem to have different emphasis. What do the authors really like to emphasize? Should the authors also discuss, in conclusion, the spatial correlation pattern between visibility and rainstorms and other issues emphasized in the abstract?

The presentation can be greatly improved. Please pay close attention on the presentation because poor presentation can hamper the readers from understanding the contents in the paper. There are numerous places requiring polishing on presentation and corrections for grammatical errors. Some examples are provided in Specific. The authors can take advantage of the editing service provided by the journal.

Figure 3 shows trends for different rain intensity. Have the authors looked into the total precipitation? What is the trend? What that trend tells us?

Figure 10 can be improved in the presentation and discussion. How significant is the correlation at each level? The statement in line 204 "indicating they were negatively correlated at low boundary layer" is not supported by Figure 10a and 10b.

Specific:

Line 33, delete "It is widely acknowledged that".

Line 46, give the full expression of CCN IN.

Line 51, use "An earlier study showed" to replace "The study shows".

Line 75-82 Data, some descriptions on quality control would be helpful.

Line 76-77, some description on MODIS data would be helpful, for example, what is the resolution of the MODIS data? How are the data used in this study?

Line 77, leave space between 200 and mm. Correct the same problem in the rest of the paper. For example, in Lines 78, 79, 94 and 95.

Line 108, Xu et al. (2016) is missing in Reference.

Line 130, delete "trends" and "extreme".

Line 112, use "the differences in the trends between" to replace "the interannual variation trend differences for".

Line 117, use "rainstorm, especially large rainstorms, have  presented a significant increase trend" to replace "rainstorm and especially large rainstorm extreme events presented significantly an increased trend".

Line 118, delete an extra comma.

Line 119-121, the sentences can be rephrased as "To compare, stations in the Tibetan Plateau (at height of >4000m), a relative clean area in China, were selected for statistical analysis of interannual variation trend of light rain frequency. The results suggest an insignificant decreasing trend for light rain frequency in the Tibetan Plateau (Fig. 3b, A)".

Line 123-126, the sentence can be rephrased as "The areas with negative trends in light rain frequency almost matched with areas with positive trends in visibility and haze frequency in EC (Fig. 4a,b and c), which are well consistent with the area of high aerosol concentrations and frequent haze events (Fig.2a,b). The light rain frequency reduction in China was closely associated with the enhancement of aerosol levels in the atmosphere (Qian et al., 2009)."

Line 123-126, what is light rain frequency? Is it the number of days with light rain in a year? What are visibility and haze frequencies? Please define them clearly in the paper.

Line 127, The sentence can be rephrased as "The areas with negative trends in light rain almost covered eastern China and a large part of China".

Line 142-143, change the phrase as "make the number of cloud droplets increase but the size of cloud droplets decrease".

Line 159-160, change the phrase as "As shown in Fig. 6b (left), in the three periods".

Line 171, use "significant increasing trends" instead.

Line 176, delete "could".

Line 197, use "PM$_{2.5}$" instead.

Lines 198, 219, 225, 226, add "s" after "concentration".

Lines 220, 225, 226, delete "s" after "droplet".

Line 224, use "These aircraft observations showed" instead.

Line 229, use "and the effects depend" to replace "depending".

Line 235, delete "trend".

Line 236, delete extra space between occurrence and more. Add a space between "with" and "an"

Line 237, add "the" before "1960"

Line 234, delete "of precipitation events" before "and".

There are various problems in the figures, their captions and annotations. The following are some examples for the authors to take into consideration.

1. Be consistent with the figure format;
2. Use the consistent fonts and font size;
3. Use correct term to label x-axis and y-axis.
4. Use capitalized words to label x-axis and y-axis;
5. Label sub-plots using letters (usually at the top, top-left, or top-right of a sub-plot);
6. Use superscripts and subscripts when necessary;
7. Provide the unit for the variable displayed if no unit, indicate with dimensionless or "(-)";
8. Indicate the unit for the color bar.
9. Remove zeros for the most insignificant digit after a decimal.
10. Add significant level (p value) on trends.
11. It is better to indicate latitude/longitude in the China maps in Figures 4 and 5.

Figure 2. Labelling sub-plots (a) and (b). Capitalize "pressure" for the label for the y-axis in Figure 2a. It should be " Pressure (hPa)" so to leave a space between "pressure" and its unit. In the caption, wind speed should have a unit of m s$^{-1}$.  Please use correct superscripts.

Figure 3. Use the same font and font size to label sub-plots. No need for zeros after a decimal point in y-axis. Label "Precipitation" or "Rain" for the y-axis in Figure 3a. Use "Year" to label x-axis (not "date"). Add significant level (p-value).

Figure 4. Label (a), (b), (c) for the subplots. Provide the unit for haze frequency, visibility, and light rain frequency. Indicate what the dots and the background stand for. Indicate the unit for the color bar.

Figure 5. Label (a) and (b) for the subplots. Provide the unit for the trend of the rainstorm frequency. Indicate what the dots and the background stand for. Indicate the unit for the color bar.

Figure 6. Label (a), (b) and (c) for the subplots. In Figure 6a, no color is needed as this will cause confusion with Figure 6b and 6c. Good titles for each figure will help readers to understand the differences between Figure 6a and Figure 6b and 6c. Otherwise, the figure can be quite confusion. In the caption, it is better to use "the positive (negative) trend" than "the positive (negative) variability".

Figure 7. Label sub-plots at the top, top-left, or top-right. Remove zeros for the most insignificant digit after a decimal. Keep sub-plots (a) –(d) the same size.

Figure 9.  Provide the unit for the dots.

Figure 10. For the label for the c-axis, remove "index".

Figure 11. What are the different marks in Figure 11a?

---

## Author Response (AR1)

**Reply to Referee 1**

**We are grateful to the referee for the encouraging comments and careful revisions, which helped to improve the quality of our paper. In the following we quoted each review question in the square brackets and added our response after each paragraph.**

*[The paper can be improved with more in-depth discussion. In the introduction, the authors can write a more comprehensive literature review on the precipitation-aerosol relationship, such as, what are controversy issues, what are the research gaps and what are the possible underlying mechanisms for various processes. Then the review can lead to what the authors will address in this paper. The conclusions and abstract seem to have different emphasis. What do the authors really like to emphasize? Should the authors also discuss, in conclusion, the spatial correlation pattern between visibility and rainstorms and other issues emphasized in the abstract?]*

**Reply 1:** Many thanks to the reviewer for the great suggestions. Following the suggestions, we have substantially improved the manuscript with more in-depth discussion with modifying the introduction about a more comprehensive literature review on the precipitation-aerosol relationship and highlighting our study results in both conclusions and abstract.

Please find the detailed revisions in the uploaded marked-up manuscript version with track changes.

*[The presentation can be greatly improved. Please pay close attention on the presentation because poor presentation can hamper the readers from understanding the contents in the paper. There are numerous places requiring polishing on presentation and corrections for grammatical errors. Some examples are provided in Specific. The authors can take advantage of the editing service provided by the journal.]*

**Reply 2:** We are grateful to the referee for encouraging comments. We have greatly improved the presentation with rewriting the sentences and correcting the grammatical errors in the revised manuscript. Please find the detailed revisions in the uploaded marked-up manuscript version with track changes.

*[Figure 3 shows trends for different rain intensity. Have the authors looked into the total precipitation? What is the trend? What that trend tells us?]*

**Reply 3:** Thanks for the suggestions. We have looked into the total precipitation averaged over Eastern China. The trend in the interannual variations of the total precipitation from 1961 to 2010 is insignificant, which indicates that the impact of aerosols on precipitation could be complicated by different rain intensity.

 *[Figure 10 can be improved in the presentation and discussion. How significant is the correlation at each level?*

*The statement in line 204 "indicating they were negatively correlated at low boundary layer" is not supported by*

*Figure 10a and 10b.]*

**Reply 4:** Following the referee's suggestion, we have modify Fig. 10 and the corresponding text in the revised manuscript as follows:

To reveal the relationship between aerosols and atmospheric vertical thermal structure, the correlation between surface $PM_{2.5}$ concentration and atmospheric thermal structure in both polluted and clean areas in July, 2013 was investigated (Fig. 10). The stations of Changsha and Hongjia located in Hunan and Zhejiang provinces in EC respectively were selected to represent the less light rain region while those of Linzhi and Dingriin of Tibet were selected to represent the high-frequency light rain region. The correlation coefficient profiles between the observed surface daily $PM_{2.5}$ concentration and atmospheric temperature profiles derived from high-resolution L-band sounding were calculated. The correlations at Changsha and Hongjia stations (Figs.10a-b) show that the correlation between $PM_{2.5}$ and temperature profiles presented an "inverse phase" pattern,  reflecting the high aerosol concentrations in a thermal stable structure similar to temperature inversion layers with "cold at low-layer and warm at upper-layer" in the eastern China. On the contrary, the correlations in Linzhi and Dingri stations in the Tibetan Plateau (Fig. 10c-d) indicate that an unstable atmospheric structure with "warm at low-layer and cold at upper-layer" for a favorable condition for the occurrence and development of convection and light rain events in the Tibetan region.

.

*[Line 33, delete "It is widely acknowledged that". ]*

**Reply 5:** it has been deleted in the revised manuscript.

*[Line 46, give the full expression of CCN IN.]*

**Reply 6:**  The full expressions of CCN and IN have been given with "cloud condensation nuclei" and "ice nuclei" in the revised muanscript.

*[Line 51, use "An earlier study showed" to replace "The study shows".]*

**Reply 7:**  It has been changed. .

*[Line 75-82 Data, some descriptions on quality control would be helpful.]*

**Reply 8:** The precipitation data are archived at the China Meteorological Administration(CMA) with the conventional quality control of global climate data.

*[Line 76-77, some description on MODIS data would be helpful, for example, what is the resolution of the MODIS data? How are the data used in this study?]*

**Reply 9:** because the MODIS aerosol products are not used in the result analysis of revised manuscript, we have deleted the sentence "annual average AOD data in 2001-2010 from Moderate Resolution Imaging Spectroradiometer (MODIS)" at the beginning of Section 2. Data. Therefore, we have not given any information on MODIS data in the revised manuscript..

*[leave space between 200 and mm. Correct the same problem in the rest of the paper. For example, in Lines 78, 79, 94 and 95.]*

**Reply 10**: Thank the referee for careful review. It has been corrected.

*[Line 108, Xu et al. (2016) is missing in Reference]*

**Reply 11:** we have added the following ACP-paper into References:

Xu, X., Zhao, T., Liu, F., Gong, S. L., Kristovich, D., Lu, C., Guo, Y., Cheng, X., Wang, Y., and Ding, G.: Climate modulation of the Tibetan Plateau on haze in China, Atmos. Chem. Phys., 16, 1365-1375, doi:10.5194/acp-16-1365-2016, 2016.

*[Line 130, delete "trends" and "extreme".*

*Line 112, use "the differences in the trends between" to replace "the interannual variation trend differences for".*

*Line 117, use "rainstorm, especially large rainstorms, have presented a significant increase trend" to replace "rainstorm and especially large rainstorm extreme events presented significantly an increased trend".*

*Line 118, delete an extra comma.]*

**Reply 12:** The careful reviews are greatly appreciated. All the errors have been corrected in the revised manuscript.

*[Line 130, delete "trends" and "extreme".]*

**Reply 13:** The have been deleted.

*[Line 112, use "the differences in the trends between" to replace "the interannual variation trend differences*

*for". ]*

**Reply 14:** It has been done in the revised manuscript.

*[Line 117, use "rainstorm, especially large rainstorms, have presented a significant increase trend" to replace*

*"rainstorm and especially large rainstorm extreme events presented significantly an increased trend".]*

**Reply 15:** Following the referee's comments, it has been revised.

*[Line 118, delete an extra comma.]*

**Reply 16:** It has been deleted.

*[Line 123-126, the sentence can be rephrased as "The areas with negative trends in light rain frequency almost*

*matched with areas with positive trends in visibility and haze frequency in EC (Fig. 4a,b and c), which are well*

*consistent with the area of high aerosol concentrations and frequent haze events (Fig.2a,b). The light rain*

*frequency reduction in China was closely associated with the enhancement of aerosol levels in the atmosphere*

*(Qian et al., 2009)." ]*

**Reply 17 :** Following the suggestion, the lines 123-126 have be rephrased as "The areas with negative trends in light rain frequency almost matched with areas with positive trends in visibility and haze frequency in EC (Fig.

4a,b and c), which are well consistent with the area of high aerosol concentrations and frequent haze events (Fig.2a,b). The light rain frequency reduction in China was closely associated with the enhancement of aerosol levels in the atmosphere (Qian et al., 2009)." in the revised manuscript.

*[Line 123-126, what is light rain frequency? Is it the number of days with light rain in a year? What are*

*visibility and haze frequencies? Please define them clearly in the paper.]*

**Reply 18:** It has been clarified with " The light rain frequency is the number of days with light rain in a year; and the visibility is in unit of km, haze frequency is the number of days with haze" in the revised manuscript.

*[Line 127, The sentence can be rephrased as "The areas with negative trends in light rain almost covered eastern*

*China and a large part of China".]*

**Reply 19:** The sentence has been rephrased as "The areas with negative trends in light rain almost covered eastern China and a large part of China" following the referee's suggestion. .

*[Line 142-143, change the phrase as "make the number of cloud droplets increase but the size of cloud droplets*

*decrease".]*

**Reply 20:**   Thanks for the suggestion. Following the referee's suggestion, it has been changed.

*[Line 159-160, change the phrase as "As shown in Fig. 6b (left), in the three periods". ]*

**Reply 21:**   It has been changed.

*[Line 171, use "significant increasing trends" instead. ]*

**Reply 22:**   It has been done.

*[Line 176, delete "could".]*

**Reply 23:** Thank the referee for the kind suggestions. It has been done.

*[Line 197, use "PM2.5" instead.]*

**Reply 24:**   It has been done.

*[Lines 198, 219, 225, 226, add "s" after "concentration".]*

**Reply 25:** It has been done.

*[Lines 220, 225, 226, delete "s" after "droplet". ]*

**Reply 26:**   It has been done.

*[Line 224, use "These aircraft observations showed" instead. ]*

**Reply 27:** It has been changed in the revised manuscript.

*[Line 229, use "and the effects depend" to replace "depending". ]*

**Reply 28:** It has been done    in the revised manuscript.

*[Line 235, delete "trend". ]*

**Reply 29:** The "trend " has been deleted in the revised manuscript.

*[Line 236, delete extra space between occurrence and more. Add a space between "with" and "an"]*

**Reply 30:** It has been done in the revised manuscript.

*[Line 237, add "the" before "1960"]*

**Reply 31:** It has been changed    in the revised manuscript.

*[Line 234, delete "of precipitation events" before "and". ]*

**Reply 32:** It has been deleted in the revised manuscript.

*[There are various problems in the figures, their captions and annotations. The following are some examples for*

*the authors to take into consideration.*

*1. Be consistent with the figure format;*

*2. Use the consistent fonts and font size;*

*3. Use correct term to label x-axis and y-axis.*

*4. Use capitalized words to label x-axis and y-axis;*

*5. Label sub-plots using letters (usually at the top, top-left, or top-right of a sub-plot);*

*6. Use superscripts and subscripts when necessary;*

*7. Provide the unit for the variable displayed if no unit, indicate with dimensionless or "(-)";*

*8. Indicate the unit for the color bar.*

*9. Remove zeros for the most insignificant digit after a decimal.*

*10. Add significant level (p value) on trends.*

*11. It is better to indicate latitude/longitude in the China maps in Figures 4 and 5.   ]*

**Reply 33:** We are very grateful to the referee for the encouraging comments and careful revisions. All the mentioned problems in the figures, their captions and annotations have been corrected in the revised manuscript.

All the figures have been redrawn following the referee's suggetsions..

*[Figure 2. Labelling sub-plots (a) and (b). Capitalize "pressure" for the label for the y-axis in Figure 2a. It*

*should be " Pressure (hPa)" so to leave a space between "pressure" and its unit. In the caption, wind speed*

*should have a unit of m s-1. Please use correct superscripts.]*

**Reply 34:** Following the referee's suggestion, Figure 2 has been modified as follows:

[Figure]

*[Figure 3. Use the same font and font size to label sub-plots. No need for zeros after a decimal point in y-axis.*

*Label "Precipitation" or "Rain" for the y-axis in Figure 3a. Use "Year" to label x-axis (not "date"). Add*

*significant level (p-value).]*

**Reply 35:** Following the suggestion, has Fig. 3 suggested been modified in the revised manuscript.

*[Figure 4. Label (a), (b), (c) for the subplots. Provide the unit for haze frequency, visibility, and light rain*

*frequency. Indicate what the dots and the background stand for. Indicate the unit for the color bar.]*

**Reply 36:** We have changed Figure 4. Label (a), (b), (c) for the subplots. In the revised caption of Fig. 4, we have added "haze frequency (day), visibility (km), and light rain frequency (day ). The dots stand for observation sites with shading by the variation trends, the background presents the terrain height in mainland China".

*[Figure 5. Label (a) and (b) for the subplots. Provide the unit for the trend of the rainstorm frequency. Indicate*

*what the dots and the background stand for. Indicate the unit for the color bar.]*

**Reply 37:** We have redrawn Fig. 4 as follows and revised the caption as "the trend (day per year) of the rainstorm frequency and the background (shaded colors) stands for the terrain height (m) in mainland China

[Figure]

[Figure]

*[Figure 6. Label (a), (b) and (c) for the subplots. In Figure 6a, no color is needed as this will cause confusion*

*with Figure 6b and 6c. Good titles for each figure will help readers to understand the differences between Figure*

*6a and Figure 6b and 6c. Otherwise, the figure can be quite confusion. In the caption, it is better to use "the*

*positive (negative) trend" than "the positive (negative) variability".]*

**Reply 38:** We have redrawn Fig. 5 as follows, and following the suggestiuon, the caption has been revised.

[Figure]

*[Figure 7. Label sub-plots at the top, top-left, or top-right. Remove zeros for the most insignificant digit after a decimal. Keep sub-plots (a) –(d) the same size.]*

**Reply 39:** Following the referee's suggestion, in the revised manuscript, Figs. 7a -7f have been redrawn as follows:

[Figure]

[Figure]

[Figure]

[Figure]

[Figure]

*[Figure 9. Provide the unit for the dots.]*

**Reply 40:** The revised caption of Figure 9 has provided the unit of day for the dots.

*[Figure 10. For the label for the c-axis, remove "index". ]*

**Reply 41:** It has been removed    in the revised manuscript.

*[Figure 11. What are the different marks in Figure 11a? ]*

**Reply 42:** In the revised caption of Figure 11a, we have added "the different marks represent the different flights"**.**

**Reply to Referee 2**

**We are grateful to the referee for the encouraging comments and careful revisions which helped to**

**improve the quality of our paper. In the following we quoted each review question in the square brackets**

**and added our response after each paragraph.**

*[The manuscript is thorough, clear, compelling, very well written, and presents the results*

*with good figures and tables. I recommend publication after attending to the following detailed comments.]*

**Reply 1:** the referee's encouraging comments are great appreciated. We have revised our manuscript following the detailed \ comments of referee.

*[Line 71: Please give references for the previous investigations of this issue primarily*

*focused on limited cases (references).]*

**Reply 2:** Thanks for the suggestion. Accordingly, we have given the following references in the revised manuscript:

Li, Z., Niu, F., Fan, J., Liu, Y., Rosenfeld, D. and Ding, Y.: Long-term impacts of aerosols on the vertical development of clouds and precipitation, Nat. Geosci. 4, 888–894, 2011.

Rosenfeld, D., Dai, J., Yu, X., Yao, Z., Xu, X., Yang, X. and Du, C.: Inverse relations between amounts of air pollution and orographic precipitation, Science 315,1396-1398, 2007.

Zhao, T. Liu, D., Zheng, X., Yang, L., Gu, X., Hu, J., Shu, Z., Chang, J., Wu, X.: Revealed variations of air quality in industrial development over a remote plateau of Southwest China: an application of atmospheric visibility data, Meteorol Atmos Phys,    doi:10.1007/s00703-016-0492-7, 2016.

*[Please give a website or reference for MODIS data.]*

**Reply 3:** because the MODIS aerosol products are not used in the result analysis of revised manuscript, we have deleted the sentence "annual average AOD data in 2001-2010 from Moderate Resolution Imaging

Spectroradiometer (MODIS)" at the beginning of Section 2. Data. Therefore, we have not given any information on MODIS data in the revised manuscript..

*[For all of Chinese map, if the author can use LambertEqualArea projection*

*(http://ncl.ucar.edu/Applications/maponly.shtml), that would be nice.]*

**Reply 4:** Thanks for the suggestion, In order to more clearly present the regional distributions of our results over mainland of China, we have used cylindrical map projection for all of Chinese map..

*[Fig. 10: Please restrict the four panels in the same size. ]*

**Reply 5:** Following the referee's suggestion, we have restricted the four panels of Fig. 10 in the same size. Please see the modified Fig. 10 in the revised manuscript.

.

[revised manuscript text omitted]

It is noteworthy that the negative trend areas of light rain  covered  all the sites in EC

(Fig. 4c). This might be also closely related with temporal and spatial variatio trends of summerEast

Asian summer monsoons activity which offered a suitable dynamic background for the effect of aerosols on clouds and precipitation. Figure 5a shows that the spatial distribution of the trends in rainstorm frequency variability was "southern positive and northern negative" in summer during 1961-2010, while the correlations between visibility and low-level cloud amount were distributed with the "northern positive and    southern negative" pattern in EC during 1961-2010 in EC (Fig.5b), indicating that the effect of aerosols on summer convective precipitation was more obvious in southern part than that in northern part of EC.

There were obvious differences in the interdecadal precipitation decreasing change rate of the various precipitation intensities in the EC region (Fig.6a), where the . In this region, the summer interdecadal precipitation variability revealed the influence features of aerosols on clouds and precipitation process, that negative variability stations of light rain made up the majority (about 87.6%), the positive variability stations of moderate rain were approximately equal to the negative ones (about 51%), the positive variability stations of heavy large rain (about 71.3%) were much more than the negative ones indicating the reverse trend. The positive variability stations of rainstorm with daily precipit ation  >50mm, including catastrophic rainstorm over 100mm occupied obvious majority (about 78.9%). In China, in the recent decades, tThe rapid increase of the anthropogenic aerosol particles in the atmosphere may not only make the cloud droplet number concentration increased but the size of cloud droplet decreased, thus changing the life time of the cloud and suppressing the precipitation, especially for the light rain (Qian et al., 2009). but also enhance the rainstorm precipitation with more frequent events in EC.

As mentioned above, the light rain frequency reduced significantly; the moderate rain frequency changed unobvious; heavy rain increased relatively obvious, rainstorm and catastrophic rainstorm increased significantly obvious in eastern China, indicating the obvious anomalous change characteristicss of regional precipitation from less light rain changed to more heavy rain and even the catastrophic rainstorm along with the frequent haze trend pollution in EC.

Although severe precipitation events mainly depended on dynamical and thermodynamic processes and water vapor source in the atmospheric circulation and deep convective activitye, aerosol's "Albrecht effect" considered that increasedwith increasing cloud droplet concentrations and decreased decreasing cloud droplet size influenced by aerosol would could suppress cloud precipitation process and extend cloud maintenance life time. The extension of the cloud life time might save the potential that triggering the abnormal severe precipitation extreme events when the cloud droplets coagulation condition was mature. This mechanism could partly explain the significant light rain reduction trend (Fig. 6a) and the spatial consistency indicating the precipitation rate transformation characteristics degrading of from light rain to heavy rain or severe extreme precipitation extreme events (blue arrows in Fig. 6a) in the polluted eastern ChinaEC region. Moreover, in this research, in order to fFurther-morewe comparatively analyzed the effects of aerosol pollution in hazy EC on extreme precipitation events, we selected the region of Tibetan Plateau (west of 110°E, south of 40°N) of Tibetan Plateau (TP), a relative clean area in western China was selected, as the reference area to comparatively analyze   the effects of aerosol pollution in hazy EC on regional precipitation change. and . we calculated percentages of sites with negative frequency variability trends of light rain and positive trends of rainstorm events in total sites with the negative and positive and negative trends in visibilityhaze over the EC and TP regions duringin the three different interdecadal periods (1961-1980, 1971-2000, 1981-2010) in the (Fig. 6b) east and westEC and TP regions. As Dueing past more then 5 decades, could be seen from Fig. 6b (left), in the three stages, tThe positive variability of light rain and reainstorm were was steady a declining trended and augmented, receptively in the polluted EC, while the negative variability was an increasing trend in EC, while there were no obvious positive or and negative variability trend of light rain and rainstorm in the clean TP region Tibetan Plateau (double-headed arrows inright panel of Fig. 6b right).

**6. interannual variability anomalies between atmospheric visibility and precipitation anomalies**

[revised manuscript text omitted]